# Epicardial Adipose Tissue and Renal Disease

**DOI:** 10.3390/jcm8030299

**Published:** 2019-03-02

**Authors:** Narothama Reddy Aeddula, Wisit Cheungpasitporn, Charat Thongprayoon, Samata Pathireddy

**Affiliations:** 1Division of Nephrology, Department of Medicine, Deaconess Health System, 600 Mary street, Evansville, IN 47710, USA; 2Department of Medicine, Indiana University School of Medicine, Evansville, IN 47708, USA; 3Division of Nephrology, Department of Medicine, University of Mississippi Medical Center, Jackson, MS 39216, USA; wcheungpasitporn@gmail.com; 4Division of Nephrology and Hypertension, Department of Medicine, Mayo Clinic, Rochester, MN 55905, USA; charat.thongprayoon@gmail.com; 5Division of Internal Medicine, Department of Medicine, Deaconess Health System, Evansville, IN 47747, USA; drspathireddy@gmail.com

**Keywords:** Epicardial Adipose Tissue, Renal disease, Chronic Kidney disease, Inflammation, Coronary artery calcification, Cardiovascular risk

## Abstract

Epicardial adipose tissue (EAT) is derived from splanchnic mesoderm, localized anatomically between the myocardium and pericardial visceral layer, and surrounds the coronary arteries. Being a metabolically active organ, EAT secretes numerous cytokines, which moderate cardiovascular morphology and function. Through its paracrine and vasocrine secretions, EAT may play a prominent role in modulating cardiac function. EAT protects the heart in normal physiological conditions by secreting a variety of adipokines with anti-atherosclerotic properties, and in contrast, secretes inflammatory molecules in pathologic conditions that may play a dynamic role in the pathogenesis of cardiovascular diseases by promoting atherosclerosis. Considerable research has been focused on comparing the anatomical and biochemical features of EAT in healthy people, and a variety of disease conditions such as cardiovascular diseases and renal diseases. The global cardiovascular morbidity and mortality in renal disease are high, and there is a paucity of concrete evidence and societal guidelines to detect early cardiovascular disease (CVD) in this group of patients. Here we performed a clinical review on the existing evidence and knowledge on EAT in patients with renal disease, to evaluate its application as a reliable, early, noninvasive biomarker and indicator for CVD, and to assess its significance in cardiovascular risk stratification.

## 1. Introduction

Different fat compartments around the heart (see Figure 1 [1,2,3,4]) have different embryological origins and functional activity [5]. Epicardial adipose tissue (EAT), a component of the visceral fat compartment, is located between the myocardium and the pericardial visceral layer, without any separating fascia or other tissue; EAT is directly in contact with the myocardium and coronary vessels [5] (see Figure 1 [1,2,3,4]). It covers 80% of the cardiac surface and accounts for approximately 20% of the total cardiac weight [6]. EAT is a derivative of splanchnopleuric mesoderm and is comprised of different cell types including adipocytes, nervous and nodal tissue, inflammatory, stromal, and immune cells [7].

Like other white adipose tissues, EAT has an endocrine function with the ability to secrete hormones and inflammatory cytokines [5,7]. Evidence suggests that EAT plays a multifaceted role in essential physiological functions of the heart such as regulation of local fatty acid homeostasis, providing a local energy source, angiogenesis, protecting cardiomyocytes from lipotoxicity by reducing the influx of high free fatty acids, coronary artery remodeling, safeguarding the heart against hypothermia, and buffering of the coronary arteries against torsion induced by myocardial contraction and arterial pulse wave [4,8,9] (see Figure 1 [1,2,3,4]). Furthermore, the adipocytokines secreted from EAT such as adiponectin, apelin, adrenomedullin, leptin, and omentin have cardioprotective effects [3]. The expression and secretion of proinflammatory cytokines, such as interleukin-6 (IL-6), interleukin-1β (IL-1β), monocyte chemoattractant protein 1 (MCP-1), and tumor necrosis factor-α (TNF-α), are higher in the EAT of patients with coronary heart disease than in subcutaneous fat of the same individuals, and the pathological enlargement of EAT correlates significantly with increased cardiovascular disease (CVD) risk [10]. The importance of EAT concerning the inflammatory burden in CVD is established [11] with the paracrine and vasocrine secretion of pro-inflammatory adipokines. Adipokines are thought to directly contribute to the myocardial inflammation, left ventricular hypertrophy (LVH), coronary artery disease (CAD), and myocardial dysfunction through the generation of local reactive oxygen species, creating a pro-atherogenic state, and as a direct local effect of inflammatory cytokines [10,12]. 

Chronic kidney disease (CKD) is one of the leading causes of mortality and morbidity across the globe with its incidence growing exponentially. According to different estimates, CKD has a consistent estimated global prevalence of about 11 to 13% [13]. As per the World Health Organization’s (WHO) global health estimates in 2016, kidney disease was the 12th most common cause of death worldwide, accounting for 1.18 million deaths [14]. CKD is a significant risk factor for CVD and risk increases with increased severity of the CKD. A meta-analysis of the urine albumin-to-creatinine ratio (ACR) data from 105,872 participants and urine protein dipstick measurement data from 1,128,310 participants, found that eGFR (estimated glomerular filtration rate) and albuminuria were associated with all-cause and cardiovascular mortality independently of each other and traditional cardiovascular risk factors [15]. In this study, two-fold higher mortality risk was observed in patients with eGFR 30−45 mL/min/1.73 m^2^ as compared to patients with optimal eGFR levels. Similarly, patients with an ACR of approximately 100 mg/g were found to be at two-fold higher risk of mortality as compared to those with optimal ACR levels (5 mg/g) [15]. 

Considerable evidence indicates a significant association between EAT thickness and the incidence of CVD events in CKD patients. This review summarizes the current knowledge on EAT concerning renal disease, different non-pharmacological and pharmacological modalities to reduce EAT and evaluate it as a reliable, early, new parameter for CVD risk stratification. A literature search was carried out in PubMed and Google Scholar using the various combinations of following keywords: epicardial adipose tissue, chronic kidney disease, end-stage renal disease, renal failure, coronary calcification, and cardiovascular disease. Appropriate articles/abstracts (published in the English language) identified in the search were included in the manuscript.

## 2. Pathological Roles of EAT 

Evidence indicates the involvement of cross-talk between EAT and the myocardium in cardiac disease pathogenesis [4]. It is revealed that in patients with essential hypertension and normal LV systolic function, EAT thickness is associated with left ventricular (LV) diastolic dysfunction and increased left atrial volume, independent of blood pressure, LV mass, and other risk factors [16]. In patients with preserved ejection fraction, EAT accumulation might contribute to the initial development of LV systolic dysfunction [17]. Moreover, epicardial fat volume (EFV) is identified as an independent predictor of a rapid increase in lipid-rich plaque volume [18], and acute coronary syndrome in patients with CAD [19]. 

EAT has been suggested to play a pivotal role in the initiation of atherosclerosis, mediating the secretion of bioactive molecules such as free fatty acids, and a variety of other cytokines or chemokines [1] (see Figure 1 [1,2,3,4]). By the expression profile analysis of various inflammatory molecules in EAT and subcutaneous fat isolates from patients undergoing coronary artery bypass grafting (CABG), Mazurek et al. showed significantly higher levels of chemokine MCP-1, and several inflammatory cytokines (IL-1β, IL-6, IL-6sR, and TNF-α) in EAT [7]. Furthermore, secretion of pro-inflammatory cytokines such as TNF-α, MCP-1, IL-1β, IL-6, and resistin amplified by the size changes in epicardial adipocytes along with the increased number of macrophages and T lymphocytes ultimately enhance atherogenesis [20,21].

As EAT is in direct contact with the adventitia of coronary arteries, adipocytokines from peri-adventitial EAT may pass through the coronary wall by diffusion and might promote the proliferation of vasa vasorum [22]. Corroborating this, it has been shown in the in vivo studies on pigs that the increase in intimal thickness and coronary artery remodeling could be stimulated by the external application of inflammatory cytokines such as IL-1β and MCP-1 [23,24]. In addition, EAT potentially can influence the coronary arteries through vasocrine secretion by directly releasing adipocytokines and free fatty acids (FFA) [5,11,22]. Also, few adipokines such as angiotensinogen, IL-6, MCP-1, TNF-α, and visfatin secreted by the EAT could cause inflammation, endothelial and smooth muscle cell proliferation, atherogenesis, and destabilization of atherosclerotic plaque [3]. Adiponectin is a protein hormone that moderates some of the metabolic processes, including glucose regulation and fatty acid oxidation [25]. Adiponectin, which has anti-atherogenic and anti-inflammatory properties, is secreted by EAT in physiological conditions. However, as the EAT becomes hypoxic and dysfunctional in pathologic conditions, adiponectin secretion is affected [26,27]. Notably, the activity of adiponectin and the risk of acute myocardial infarction are inversely associated [28]. 

Leptin, a hormone predominantly made by the adipose cells, is a mediator of long-term regulation of energy balance, suppressing food intake and thereby inducing weight loss. Leptin secreted by the EAT inhibits nitric oxide synthetase through protein kinase C dependent mechanisms and thereby induce endothelial dysfunction [29]. Adrenomedullin (ADR), a peptide hormone with pleiotropic effects on the vasculature, is significantly associated with EAT in certain disease conditions. Elevated ADR levels are found in CAD patients which are correlated with endothelial dysfunction [30]. Apart from this, vitamin D deficiency has been associated with increased expression of inflammatory markers in EAT among animal models [31]. However, it is still unknown if this mechanism applies to humans [32]. Collectively, EAT hypertrophy is observed under various pathological conditions, and the adipocytokines secreted by the infiltrated immune cells in the EAT could potentially induce atherosclerotic cardiac disease.

## 3. Measurement of EAT 

Given the widespread availability of non- invasive imaging, EAT measurement has been increasingly performed in the general population, patients with cardiovascular disease, obesity, and diabetes, as it is a reliable predictor of cardiovascular risk, independent of traditional risk factors and other fat depots [5]. Echocardiography, computed tomography (CT), or magnetic resonance imaging (MRI) are the three methods that are used to measure and quantify EAT [5,8]. Various advantages and disadvantages of these three modalities are given in Table 1 [1,33,34,35,36,37,38,39]. Echocardiography is a simple, inexpensive, and readily available in health care facilities; however, with high intra- and inter-observer variability, inability to quantify the epicardial fat, and at times overestimation of EAT are some of its significant drawbacks. Precise measurements of EAT could be made with MRI. Despite MRI being a gold standard for EAT measurements, in everyday practice, it is hindered by several drawbacks such as high cost, less availability, and contraindications in patients with pacemakers and implants. Cardiac CT provides better EAT assessment with the highest specificity and sensitivity. An ability to obtain a coronary artery calcification (CAC) score with the cardiac CT improves the cardiovascular risk assessment [26]. Cardiac CT involves only a small radiation dose (1 mSv). 

## 4. Reference Values of EAT 

Considering the growing body of evidence indicating the profound role of EAT in the pathophysiology of CVD, it is imperative to establish the reference ranges of EAT to differentiate or identify the population at risk. Furthermore, to establish EAT as a diagnostic marker, defining the reference ranges becomes a fundamental prerequisite. To this extent, some studies have set forth to evaluate the EAT thickness in different subject groups. Through the histological analysis of 200 human hearts, Schejbal has found the mean EAT thickness of the right ventricle (sharp heart edge close to the bases) as 4.12 mm [40]. Similar observations were made by Flüchter et al. who estimated the mean right ventricular EAT thickness using cardiac MRI as 4.1 ± 1.1 mm in 28 healthy subjects. In the same study, the EAT thickness was found to be less (3.2 ± 1.2 mm) in congestive heart failure patients [41]. 

Of the two analyses carried out on the Framingham Heart MDCT (multi-detector CT) Study participants, Rosito GA et al. determined the EAT thickness as 110 cm^3^ and 137 cm^3^ in men and women respectively, and Mahabadi AA et al. estimated the mean EAT volume (of the overall subjects included in the study) as 124 ± 50 cm^3^ [42,43]. Furthermore, higher EAT thicknesses were observed in untreated hypertensive patients with grade 1 (impaired relaxation pattern; 7.2 ± 2.4 mm) and grade 2 (pseudo-normal pattern diastolic dysfunction; 7.7 ± 2.3 mm) diastolic dysfunction as compared to subjects with normal diastolic function (5.4 ± 1.8 mm) [15]. Demir et al. estimated the EAT thickness of 6.0 ± 2.0 mm in patients with metabolic syndrome as compared to 4.0 ± 1.0 mm in control group patients (*p* < 0.001) [44]. In a study involving 950 Indian subjects, EFV was significantly higher in patients with non-obstructive CAD (68.67 ± 29.18 mL) and obstructive CAD (82.872 ± 32.32 mL) as compared to subjects with no CAD (56.73 ± 27.63 mL) (*p* < 0.001) [45].

From the existing evidence, it can be considered that EAT thicknesses > 5 mm, or volume > 125 mL, or 68 mL/m^2^ as abnormal [35]. However, an extensive review of the literature with a systematic approach corroborated with ample clinical experience of the experts from the field is the need of the hour to establish internationally accepted and reliable reference values for EAT.

## 5. EAT in Chronic Kidney Disease (CKD) and End-Stage Renal Disease (ESRD)

Chronic kidney disease involves chronic inflammation that is exemplified by the presence of increased levels of inflammatory markers such as C-reactive protein (CRP), IL-6, and TNF-α in the circulation [6]. Higher inflammation and oxidative stress are known to play a pivotal role in the development of atherosclerosis in patients with CKD [6]. A growing body of evidence indicates a strong association between increased EAT thickness, inflammatory markers, and CAC in patients with CKD. In a Japanese study on 110 early CKD and 165 non-CKD patients by Nakanishi et al. cardiac CT scan showed significantly increased EAT volume and vulnerable plaque in CKD patients [46]. 

Several studies have demonstrated that the EAT thickness is high in hemodialysis or peritoneal dialysis patients than the healthy subjects (see Table 2 [47,48,49,50,51,52,53]). In a study by Sheng et al. n 120 patients with CKD and 30 healthy subjects, cardiac CT showed higher EAT volume in stage 4–5 D CKD group patients compared with the control group; however, EAT volume was similar between CKD 3 and control group [54]. Apart from this, the EFV was significantly higher in the peritoneal dialysis group than in the hemodialysis group, CAD group compared with no CAD group, and in the diabetes group than in no diabetes group. Significant research has been dedicated to understanding the association of EAT thickness in hemodialysis patients with different parameters. In a cross-sectional study on 43 hemodialysis patients and 30 healthy subjects, the neutrophil-to-lymphocyte ratio (a novel inflammatory marker) was found to be an independent predictor of EAT in hemodialysis patients [55]. In another cross-sectional study on 72 hemodialysis patients with ESRD, the activity of paraoxonase-1 (PON-1) enzyme has been shown to be inversely correlated with the EAT thickness [56]. PON-1 is the most potent high-density lipoprotein (HDL) associated antioxidant enzyme whose activity has been demonstrated to be inversely correlated with oxidative stress and cardiovascular risk [56,57]. Apart from this, a strong correlation was observed between EAT thickness and ischemiamodified albumin (IMA) and myeloperoxidase (MPO) in hemodialysis patients; IMA is elevated in ESRD, MPO plays a role in inflammation and CVD [49]. An inverse correlation between EAT thickness and coenzyme Q10 levels was observed in hemodialysis patients as compared to matched healthy subjects in a study by Macunluoglu et al. (*p* < 0.05) [58]. 

## 6. EAT in Renal Transplantation

Limited literature is available on the EAT in kidney transplant patients. In a post hoc analysis on 98 kidney transplant patients, the impact of the epicardial fat gain on ventricular mass after kidney transplantation could not be positively confirmed [59]. Similar to hemodialysis patients, PON-1 activity was found to be inversely correlated with EAT thickness in renal transplant patients [60]. When EAT changes were compared between hemodialysis patients and kidney transplant patients, EAT measurements were significantly higher in hemodialysis patients than in the kidney transplant patients, and the EAT measurements of the kidney transplant patients were not statistically different from the healthy subjects [61]. This suggests that kidney transplantation could reduce EAT as the kidney transplant patients in this study were previously on hemodialysis. 

## 7. Microalbuminuria and EAT

Studies demonstrated higher EFV in microalbuminuria patients. In a study on essential hypertensive patients, mean EAT thickness was higher in those with microalbuminuria as compared to the normoalbuminuric patients with significant positive correlations between EAT and LV mass and LV mass index [62]. Akbas et al. made similar observations in type 2 diabetes patients where EAT thickness was significantly high in macro- albuminuric or micro-albuminuric patients than normoalbuminuric patients [63]. Furthermore, EAT was identified as an independent predictor of increased albuminuria in this study. Likewise larger EAT volumes were observed in type 1 diabetes patients with a history of an albumin excretion rate ≥300 mg/dL [64]. 

## 8. EAT and Coronary Artery Calcification in Renal Disease

Cardiovascular diseases are the most common cause of morbidity and mortality in CKD patients [65]. CAC indicates the presence of coronary atherosclerosis [66], and it usually correlates with increased cardiovascular risk and poor outcomes mainly in patients with CKD [67,68,69,70]. EAT measurement is generally used to confirm the association with CAC; it characterizes the presence of high-risk coronary artery plaque [69,71]. It is notable that the volume of EAT is eligible as a marker of coronary atherosclerosis even with a coronary artery calcium score of zero on coronary CT angiography [72]. In a large meta-analysis by Nerlekar et al., increased EAT and the presence of high-risk plaque are significantly correlated [73]. In the prospective EVASCAN (EVAluation of CT SCANner) study that investigated the association of EAT thickness with the extent of angiographic CAD, EAT thickness of left ventricle wall was associated with CAD and was identified as an independent predictor of CAD [74]. In the Cardiometabolic risk, Epicardial fat, and Subclinical Atherosclerosis Registry (CAESAR) study participants, increase in EAT and the extent of calcium in the coronary artery are significantly correlated [75]. In a study on the normotensive patients with autosomal dominant polycystic kidney disease (ADPKD), EAT thickness, and carotid intima-media thickness (CIMT) was higher than the control subjects [76], In addition, a statistically significant positive association was observed between EAT thickness and CIMT [76]. A robust association was observed between CAC and EFV in pre-dialysis stage 3–4 CKD patients, and the EFV is greater in individuals with metabolic syndrome, increased insulin resistance, or diabetes mellitus [77]. Through a retrospective analysis on 529 patients with high risk of coronary atherosclerotic nephropathy, Zuo et al. have identified higher EAT thickness in patients with atherosclerosis [78]. The authors suggest that measuring EAT volume could predict the occurrence and development of coronary atherosclerosis in the future [78]. The study also found a positive correlation between EAT volume and coronary artery calcification, BMI and number of coronary lesions in patients with nephropathy [78]. 

When the EAT and CAC score was investigated in diabetic, nondiabetic ESRD patients, and healthy subjects, total CAC scores and EAT measurements were significantly higher in diabetic ESRD patients [79]. A statistically significant relationship was observed between EAT and CAC scores in ESRD patients (*p* < 0.0001) [79]. Furthermore, statistically significant higher EAT measurements were observed in both diabetic and non-diabetic ESRD patients when compared to healthy subjects (*p* < 0.01 for healthy subjects vs. ESRD patients). In a study that investigated the relationship between EAT and surrogate cardiovascular markers in patients on maintenance hemodialysis, EFV was significantly higher in patients with higher CAC scores (>10), and it correlated with atherosclerosis, arterial stiffness, and the presence of CAC [80]. However, in a study of 93 patients on long-term hemodialysis, save for the subjects younger than 55 years of age, the EFV was not significantly correlated to the CAC [81]. Taken together, information from most of the studies suggest a strong correlation between EAT thickness and CAC in renal disease patients.

## 9. Clinical Significance of EAT in Renal Disease

In CKD patients, EAT assessment could be a reliable parameter for cardiovascular risk assessment [69]. Inflammation in CKD patients is remarkable when compared to the general population as malnutrition adds onto the already existing high inflammatory state of these patients; as a result, inflammation caused by adipose tissue is more relevant in CKD patients [52,69]. Despite some limitations, the study by Cordeiro et al. provides valuable insights into the association between EAT and CVD risk factors and CV events in CKD patients [82]. In this single-center, cross-sectional, prospective observational study on 227 non-dialyzed stage 3–5 CKD patients, an increase in EAT thickness was associated with severity of CVD; as the EAT thickness increased, the high prevalence of LV hypertrophy and myocardial ischemia was observed along with high CAC score. Furthermore, elevated EAT thickness was associated with poor CVD prognosis. Another important observation from this study is that higher EAT was associated with an increased risk of cardiovascular events independently of visceral (abdominal) fat and other potential confounders [82]. Notably, the authors indicate that EAT does not add any meaningful prognostic power beyond visceral fat and CKD related risk factors. Apart from this, in a 6.1 year follow-up, prospective study on 200 type 2 diabetes patients with elevated urinary albumin excretion rate, EAT was significantly (*p* = 0.029) associated with CVD and mortality even after accounting for traditional CVD risk factors [83]. Moreover, it is suggested that EAT is not a cardiovascular risk indicator in hemodialysis patients without diabetes mellitus and hypertension [84]. However, in a study on 177 outpatients at intermediate risk for CAD, EAT volume was suggested to provide incremental predictive value for future cardiac events [85]. Likewise, in a study by Reinhardt et al., in individuals with early adult onset T2DM, EFV was suggested as a risk factor for heart disease, and in individuals with youth-onset T2DM, it is indicated as a predictor of decreased kidney function [86]. Authors of the study commented that, while EFV might influence renal disease in youth, and renal disease is a precursor to macrovascular disease, it is more likely that EFV is a marker of metabolic syndrome predisposing to both conditions [86]. Metabolic syndrome is linked to epicardial fat storage and EAT might just be a marker of metabolic syndrome or exacerbate the risk of metabolic syndrome on kidney disease [86]. Collectively, it seems that EAT has a prominent role in precipitating CVD events, and these observations deserve thorough examination and consideration in clinical practice, and must be extensively studied further through high-quality clinical studies.

In a post-hoc analysis of a prospective study on 59 hemodialysis patients, EAT correlated significantly with cardiovascular calcification in long-term HD patients [87]. In addition, EAT correlated significantly with age, BMI, CAC, and aortic calcium. In the post-hoc analysis of the Renagel in New Dialysis patients (RIND study) each 10 cm^3^ increase in EAT volume in hemodialysis patients was associated with a significant 6% increase in the risk of death during follow-up [88]. Age, BMI, CAC, and aortic calcium were also found to be positively correlated with EAT in the study. In another subgroup analysis of RIND study participants EAT progression from baseline was smaller in hemodialysis patients treated with sevelamer (calcium-free phosphate binder) than in patients treated with the calcium-containing phosphate binder [89]. In a study that investigated the relationship between EAT and MIAC (malnutrition, inflammation, atherosclerosis/calcification) syndrome in ESRD patients, total CAC score and EAT measurement were higher, EAT positively correlated with CAC score, age and BMI, and EAT was increased when MIAC components increased [52]. Overall evidence indicates that age, BMI, CIMT, CAC, and hemodialysis status as some of the factors to have a positive correlation with EAT thickness in renal disease patients and hence these factors deem specific consideration during the treatment. As considerable evidence indicates EAT as an independent predictor of CVD and mortality in renal disease patients, it is rational to acknowledge the importance of therapeutic interventions to reduce EAT.

## 10. Clinical Reduction of EAT

Strategies to decrease EAT comprise simple lifestyle modifications including diet control and exercise to medical therapy and even more complex surgical interventions. In a prospective study on 30 non-diabetic obese men with metabolic syndrome, a 3-month weight reduction plan, including diet control and exercise, resulted in a statistically significant (*p* < 0.001) reduction in EAT thickness ranging from −34% to −18% (as per measurements in various planes) in men who had >5% weight loss [90]. Interestingly, the decrement in EAT thickness correlated with improvement in insulin sensitivity index in these subjects. Likewise, in another study on 20 severely obese subjects (BMI 45 ± 5 kg/m^2^), a 6 month very low-calorie diet program resulted in 20% loss of bodyweight, 19% loss of BMI and importantly 32% loss of EAT (*p* < 0.001), as compared to baseline [91]. With weight loss, significant improvements in left ventricular mass (LVM) and diastolic function were also observed in these patients which correlated better with EAT loss rather than BMI or waist circumference reduction. 

Concerning medications, statins, and different anti-diabetes medications have been shown to reduce EAT significantly in various studies (see Table 3 for details [92,93,94,95,96,97,98,99,100]). Independent of the effects on blood cholesterol or lipoproteins, high doses of statins both lower the EAT thickness and its pro-inflammatory characteristics, and thus reduce systemic inflammation [2]. In a study on 193 patients with aortic stenosis undergoing cardiac surgery, statin treatment was associated with significantly lower EAT thickness (*p* < 0.0001) and lower levels of EAT-secreted inflammatory mediators (*p* < 0.0001) [101]. Moreover, the study also identifies a significant correlation between EAT thickness and its inflammatory status.

Different classes of anti-diabetic drugs such as sodium glucose co-transporter-2 inhibitors (SGLT2i), glucagon-like peptide-1 receptor agonist (GLP-1RAs), and dipeptidyl peptidase-4 inhibitors (DPP-4i), have been shown to reduce the EAT thickness to a varying extent; see Table 3 for the details of the observations in various studies. The EAT reduction effect of these anti-diabetic drugs could probably be due to their impact on reducing the body weight or in some cases specifically the body fat. In a study on EAT explants, dapagliflozin, a SGLT2 i have been shown to reduce the secretion of pro-inflammatory chemokines and improve the differentiation of EAT cells [102]. When the expression of pro-inflammatory markers was quantified in EAT of CAD patients with type 2 diabetes who had undergone surgery, pioglitazone, a thiazolidinedione drug, significantly reduced the expression of IL-1β, IL-1Ra, and IL-10 [103]. In a study on 73 multi-vessel CAD patients who underwent elective bypass grafting, a combination of simvastatin (20 mg/day) and pioglitazone (15 mg or 30 mg/day) substantially reduced EAT and plasma inflammatory markers in CAD and metabolic syndrome patients [104]. 

Owing to the high sensitivity of renal disease patients to different medications, therapeutic interventions should be carefully chosen to reduce EAT thickness. The 2013 Kidney Disease: Improving Global Outcomes (KDIGO) Clinical Practice Guideline for Lipid Management in Chronic Kidney Disease recommend that statins can be used in adults aged ≥50 years with eGFR ≤ 60 mL/min/1.73 m^2^ but not treated with chronic dialysis or kidney transplantation, in adults aged ≥50 years with CKD and eGFR ≥ 60 mL/min/1.73 m^2^, in adults aged 18–49 years with CKD, but not treated with chronic dialysis or kidney transplantation, and in adult kidney transplant recipients [105]. In adults with dialysis-dependent CKD, the statins are contraindicated; however, statin treatment can be continued if a patient is already receiving statins or statin/ezetimibe combination at the time of dialysis initiation [105]. Special care should be taken while prescribing any of the anti-diabetic drugs in the CKD patients as the American Diabetes Association (ADA) recommends either dose modifications or in some cases contraindicate these drugs [106]. The ADA guidelines do not recommend canagliflozin and dapagliflozin when the eGFR is less than 45 mL/min/1.73 m^2^ and 60 mL/min/1.73 m^2^ respectively, and contraindicate dapagliflozin in patients with eGFR < 45 mL/min/1.73 m^2^ [106]. The 2012 update of the Kidney Disease Outcomes Quality Initiative (KDOQI) Clinical Practice Guideline for Diabetes and Chronic Kidney Disease do not suggest any dosage adjustment for pioglitazone [107]. For sitagliptin, the KDOQI guidelines recommend a dose of 100 mg daily when eGFR > 50 mL/min/1.73 m^2^, 50 mg daily when eGFR is 30–50 mL/min/1.73 m^2^ and 25 mg daily when eGFR < 30 mL/min/1.73 m^2^ [107]. Regarding the GLP-1RAs, the KDOQI guidelines do not recommend exenatide and liraglutide when eGFR < 30 mL/min/1.73 m^2^ and <60 mL/min/1.73 m^2^ respectively [107].

Surgical interventions involving bariatric surgery or direct surgical resection of EAT have been shown to reduce EAT. In a study on morbidly obese patients, six months after bariatric surgery, epicardial fat volume reduced from 137 ± 37 mL to 98 ± 25 mL (*p* < 0.0001), along with a substantial reduction in BMI, subcutaneous fat, and visceral abdominal fat [108]. In the case of epicardial hypertrophy, surgical resection has been shown to reduce the EAT and thereby improve cardiac function [109]. Taken together, strategies to reduce EAT thickness should initiate with lifestyle modifications followed with therapeutic intervention. Surgical procedures could be considered as the last option for EAT reduction. 

## 11. Conclusions

It has been well established that EAT plays a multifaceted role in cardiac metabolism and its importance in regulating cardiac disease pathogenesis is being accepted widely. Notably, due to its anatomic proximity to the myocardium, EAT is gaining immense attention as inflammatory molecules secreted by EAT could have a paracrine and vasocrine effect on the myocardium. As the EAT thickness could increase before the mature atherosclerotic plaque development, EAT assessment may be useful in screening and initiating treatment during the earlier stages of CVD. Also, EAT could cause coronary artery calcification and may predict its progression, which ultimately results in myocardial ischemia. 

Even though CVD poses a significant risk in ESRD patients (mortality rate due to cardiovascular events is high in ESRD patients), they are often asymptomatic due to diabetes or impaired exercise capacity [110]. Traditional screening tools lack predictive accuracy in identifying these symptoms in ESRD patients due to a variety of reasons emanating from their clinical staus [110]. Therefore, there is a quintessential need for identifying risk factor(s) or marker(s) through which future cardiac events could be diagnosed, and the patient could be provided with proper care at the right time. To this extent, we propose that measuring the EAT volume could be a marker for cardiovascular risk stratification in ESRD patients. At this point, we accede that there is a significant dearth of evidence both at the level of animal studies or on humans which strengthen our assumption. However, considerable body of current evidence indicat that CKD and ESRD patients have higher EAT volume motivates to test this hypothesis. Furthermore, to our knowledge, the cause-effect relationship between EAT volume, cardiovascular risk, and renal disease has not been established. Therefore, it also remains a fertile area for future research.

Also, future large scale trials are required to assess the prognostic significance of reducing the EAT thickness in different patient groups and especially in patients with renal disease. As a positive correlation between EAT thickness and age, BMI, CIMT, CAC and hemodialysis status has been consistently observed in renal disease patients in many independent studies, these factors could be given importance in the clinical practice. Data suggest that EAT thickness can be reduced non-pharmacologically with lifestyle modifications including low-calorie diet and anaerobic exercise. Even though low-quality studies indicate that different pharmacological interventions (mainly statins and some anti-diabetes drugs) can reduce the EAT thickness, high quality randomized, double-blind trials are required to ascertain these effects. Overall, epicardial adipose tissue could be a reliable cardiovascular clinical parameter in both CKD and non-CKD subjects that can be readily measured by non-invasive, cheap and reliable methods to stratify the patients and design the management plan. 

## Figures and Tables

**Figure 1 jcm-08-00299-f001:**
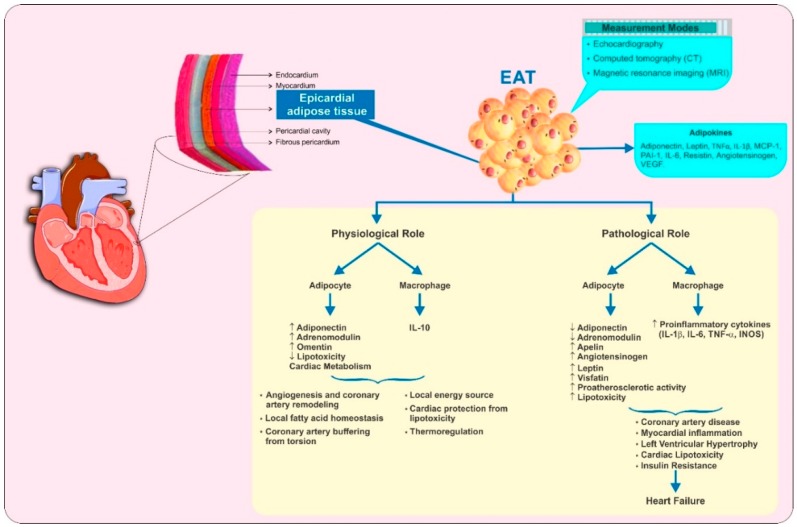
Physiological and pathological functions of epicardial adipose tissue (EAT). Schematic diagram illustrating the anatomical location of EAT, different modalities used to measure it and its various physiological and pathological functions. IL, interleukin; INOS, inducible nitric oxide synthase; MCP-1, monocyte chemoattractant protein-1; PAI-1, plasminogen activator inhibitor-1; TNF-α, tumor necrosis factor α; VEGF, vascular endothelial growth factor.

**Table 1 jcm-08-00299-t001:** Salient features of three different EAT measurement modalities.

	Advantages	Disadvantages
Echocardiography [1,33,34,35]	Widely available, non-invasive, and cost-effective Easily repeatable Independent from cardiac rhythm; Complete and accurate right ventricular function No contrast administration and no radiation	Operator-dependent Acoustic window-dependent Inconsistency in the measurement location due to spatial variations Observer-dependent No volumetric quantification Limited to one region (right ventricular free wall) Poor image quality (especially in obese individuals)
Computed Tomography (CT) [1,34,36,37,38]	The 3D technique with short total scan times and high reproducibility Convenient with shorter training time for technicians Possibility for volumetric quantification Provide image anatomy and function independent from acoustic window Combined assessment of heart, vasculature, and lung Better temporal and spatial resolution when compared to MRI Important in patients with contraindications to MRI or in centers where MRI is not available	Ionizing radiation risk Need for iodinated contrast agent which may affect renal function in patients with CKD Limited information on right ventricular structure and Function Contraindicated in pregnancy Limited in separating pericardial/epicardial adipose tissue Weight table limit for severely obese individuals
Magnetic Resonance Imaging (MRI) [1,33,34,36,37,38,39]	The gold standard for adipose tissue imaging Non-invasive, safer, 3D imaging with high reproducibility High spatial resolution and very high temporal resolution Ability to depict soft tissues with adequate contrast Can be performed with free breathing; Arrhythmias are generally not a problem Can image anatomy, function, and physiology Independent from acoustic window Possibility for volumetric quantification No need for iodinated contrast and no radiation risk	Expensive and not readily available Long total scan times Foreign matter causes local artifacts Need for long training time Several contraindications (e.g., Pacemaker, arrhythmias, claustrophobia, end-stage renal disease, implanted metallic devices) Provide limited information on lung Artifacts including motion, respiratory, cardiac motion No accommodation for severely obese individuals

CKD: Chronic kidney disease; MRI: magnetic resonance imaging.

**Table 2 jcm-08-00299-t002:** Studies evaluating the EAT thickness in patients on dialysis.

Study	Study Population	Endpoint Concerning EAT	Mode of EAT Measurement	EAT Thickness	Conclusion
**Studies on hemodialysis patients**					
Altun et al., (2014) [47]	62 hemodialysis patients and 40 healthy subjects	Evaluate the relationship between EAT thickness and CIMT	Echocardiography	Hemodialysis patients: 6.98 ± 1.67 mm; Healthy subjects: 3.84 ± 0.73 mm; *p* < 0.001	EAT thickness is higher in hemodialysis patients than in control subjects and correlated with CIMT, hemodialysis duration, and age. CIMT, hemodialysis duration and age are independent predictors of EAT thickness.
Aydin et al., (2017) [48]	60 hemodialysis patients and 84 controls, without diabetes and CVD; Prospective study	Investigate EAT thickness	Transthoracic 2D echocardiography	Hemodialysis patients: 8.0 ± 2.2 mm; Healthy subjects: 5.8 ± 1.9 mm; *p* < 0.01	Hemodialysis patients have significantly higher EAT thickness. Hemodialysis patient status is an independent predictor of EAT thickness.
Karatas et al., (2018) [49]	37 hemodialysis patients, 43 pre-dialysis patients, 31 healthy subjects	Examine the relationship of EAT thickness with IMA and MPO levels in patients with CKD	Transthoracic echocardiography	Hemodialysis patients: 6.841 ± 1.194 cm; pre-dialysis patients: 6.261 ± 1.302 cm; healthy subjects: 4.053 ± 0.611 cm; *p* < 0.001.	EAT is significantly high in the hemodialysis group as compared to pre-dialysis and healthy subjects. IMA and MPO levels are significantly high in hemodialysis subjects. EAT thickness positively correlates with IMA and MPO levels
**Studies on peritoneal dialysis patients**					
Turkmen et al., (2012) [50]	45 Peritoneal dialysis patients, 25 healthy subjects; Cross-sectional study	Investigate the relationship between EAT and CAC in peritoneal dialysis patients	MDCT	Peritoneal dialysis patients: 160 ± 77 cm^3^; Healthy subjects: 121.5 ± 37.5 cm^3^; *p* = 0.02	EAT thickness is high in peritoneal dialysis patients. EAT measurements significantly correlates with arterial calcification of the main segments of the coronary artery. Patients with BMI ≥ 30 has increased EAT than those with BMI < 30. Increased EAT positively correlated with age and BMI.
Turkmen et al., (2013) [51]	35 Peritoneal dialysis patients, 30 healthy subjects; Cross-sectional study	Evaluate PFT thickness and independent predictors of PFT in peritoneal dialysis patients	MDCT	Peritoneal dialysis patients: 170 ± 79 cm^3^; Healthy subjects: 120 ± 43 cm^3^; *p* = 0.003	PFT is strongly correlated with EAT in peritoneal dialysis patients.
**Studies involving both hemodialysis and peritoneal dialysis patients**					
Turkmen et al., (2011) [52]	80 ESRD patients either on hemodialysis or peritoneal dialysis, 27 healthy subjects; Cross-sectional study	Investigate the relationship between EAT and MIAC syndrome	MDCT	ESRD patients: 160 ± 76 cm^3^; Healthy subjects: 121.5 ± 37.5 cm^3^; *p* = 0.02	Increased EAT in ESRD patients as compared to healthy subjects. EAT measurements were higher in peritoneal dialysis patients than in hemodialysis patients EAT significantly correlated with MIAC syndrome in ESRD patients EAT significantly increased upon an increase in a number of MIAC components. EAT is positively correlated with age, BMI, and MIAC.
Erdur et al., (2013) [53]	76 ESRD patients receiving peritoneal- or hemodialysis and 42 healthy subjects; Cross-sectional study	Determine the relationship between AIP and EAT	MDCT	ESRD patients: 160 ± 76 cm^3^; Healthy subjects: 104 ± 48 cm^3^; *p* < 0.001	Compared to healthy subjects, AIP and EAT are increased in ESRD patients. Significant correlation between EAT and BMI and AIP in ESRD patients. AIP is significantly high in ESRD patients with high EAT volumes. Age and BMI are independent predictors of EAT.

AIP, atherogenic index of plasma; BMI, body mass index; CAC, coronary artery calcification; CIMT, carotid intima-media thickness; CVD, cardiovascular disease; EAT, epicardial adipose tissue; ESRD, end-stage renal disease; IMA, ischemia-modified albumin; MIAC, malnutrition, inflammation, atherosclerosis, calcification; MDCT, multi-detector computed tomography; MPO, myeloperoxidase; PFT, peri-aortic fat tissue.

**Table 3 jcm-08-00299-t003:** Evidence indicating the reduction of EAT with pharmacological intervention.

Study	Study Population; Study Duration	Pharmacological Therapy and Dose	Results	Conclusion
**Statins**				
Alexopoulos et al., (2013) [92]	420 post-menopausal women; One year	A: Atorvastatin 80 mg/day B: Pravastatin 40 mg/day	% change in EAT. A versus B*: −3.38 (−30.1 to 40.1) vs. −0.83 (−37.9 to 62.8); *p* = 0.025	Intensive lipid-lowering therapy with atorvastatin significantly reduces EAT volume.
Soucek et al., (2015) [93]	79 patients with AF who underwent pulmonary vein isolation; 3 months	A: Atorvastatin 80 mg/day B: Placebo	Median decrease in EAT volume (baseline vs. follow-up) *: A: 92.3 cm^3^ (62.0 to 133.3) vs 86.9 cm^3^ (64.1 to 124.8); *p* < 0.05. B: 81.9 cm^3^ (55.5 to 110.9) vs 81.3 cm^3^ (57.1 to 110.5), *p* = NS.	Intensive atorvastatin therapy in AF patients who underwent pulmonary vein isolation significantly decreases EAT thickness.
**Anti-diabetes medications**				
SGLT2i				
Bouchi et al., (2017) [94]	19 T2D patients; 12 weeks	Luseogliflozin 2.5 mg daily, titrated up to 5 mg daily	Median decrease in EAT volume (baseline vs. follow-up) *: 117 cm^3^ (96–136) vs. 111 cm^3^ (88–134); *p* = 0.048	Luseogliflozin significantly reduces EFV in T2D patients. EFV reduction significantly correlates with C-reactive protein reduction.
Fukuda et al., (2017) [95]	9 non-obese T2D patients; 12 weeks	Ipragliflozin 50 mg daily	Median decrease in EAT volume (baseline vs. follow-up) *: 102 (79–126) cm^3^ vs. 89 (66–109) cm^3^; *p* = 0.008	Ipragliflozin significantly reduces EFV in non-obese T2D patients. EFV reduction significantly correlates with change in BMI.
Sato et al., (2018) [96]	40 T2D patients with CAD; 6 months	A: Dapagliflozin B: Conventional treatment involving various anti-diabetes drugs	Reduction in EAT volume (A vs. B): −16.4 ± 8.3 cm^3^ vs. 4.7 ± 8.8 cm^3^; *p* = 0.01	Dapagliflozin reduces EAT volume significantly. The significant correlation observed between changes in EAT volume and the difference in body weight or TNF-α level.
Yagi et al., (2017) [97]	13 T2D patients; 6 months	Canagliflozin 100 mg	The decrease in EAT thickness (baseline vs. follow-up): 9.3 ± 2.5 vs. 7.3 ± 2.0 mm; *p* < 0.001	Canagliflozin reduces EAT thickness in T2D patients independent of its effect on blood glucose.
GLP-1 Analogues				
Dutour et al., (2016) [98]	44 obese T2D subjects uncontrolled on oral antidiabetic drugs; 26 weeks	A: Exenatide 5 µg to 10 µg b. i. d. B: Reference treatment	Reduction in EAT: A: −8.8 ± 2.1%; B: −1.2 ± 1.6%; *p* = 0.003	Exenatide significantly reduces EAT in obese T2D patients These effects mainly depend on weight loss
Iacobellis et al., (2017) [99]	95 T2D patients with BMI ≥ 27 kg/m^2^ and A1C ≤ 8%; 6 months	A: Liraglutide up to 1.8 mg s.c. o. d. + Metformin up to 1000 mg b. i. d. B: Metformin up to 1000 mg b. i. d.	Reduction in EAT (baseline vs. follow-up): A: 9.6 ± 2 mm vs. 6.2 ± 1.5 mm; *p* < 0.001 B: 7.4 ± 1.6 mm vs. 6.9 ± 1.3 mm; *p* = NS	Liraglutide added to metformin results in about 36% reduction of EAT
DPP-4i				
Lima-Martinez et al., (2106) [100]	26 T2D patients with A1C ≥ 7%; 24 weeks	A: Sitagliptin 50 mg + Metformin 1000 mg b. i. d.	Reduction in EAT (baseline vs. follow-up): A: 9.98 ± 2.63 vs. 8.10 ± 2.11 mm; *p* = 0.001	Addition of sitagliptin to metformin reduces the EAT significantly and rapidly

A1C, glycated hemoglobin; AF, atrial fibrillation; b. i. d., twice daily; BMI, body mass index; DPP-4i, dipeptidyl peptidase-4 inhibitors; EAT, epicardial adipose tissue; EFV, epicardial fat volume; GLP-1RAs, glucagon-like peptide-1 receptor agonists; o. d., once daily; NS, statistically not significant; s. c., sub-cutaneous; SGLT2i, sodium glucose co-transporter-2 inhibitors; T2D, type 2 diabetes mellitus; TNF, tumor necrosis factor; *, values represent median (range).

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
