# Peer review of "Epicardial Adipose Tissue and Renal Disease"

_jcm, 2019, doi:10.3390/jcm8030299_

Reviewer 1 Report

In this paper Aeddula et at review in detail the exiting knowledge on epicardial adipose tissue (EAT) and renal disease. Although the manuscript is clear and well organized with a complete description on the role of EAT on cardiac metabolism and disease; the link that they try to achieve between EAT and renal disease seems to be at best not well established. 

The authors propose the measurement of the EAT thickness as an early marker of cardiovascular disease (previously suggested in Russo et al J Nephrol. 2018) and discuss different strategies to reduce the EAT in patients with renal disease such as life-style changes that involve low calorie diets and anaerobic exercise. This course of treatment will impact not only EAT but all the fat depots, therefore, the particular beneficial effect of it cannot be attribute to an EAT reduction. 

In conclusion, although the authors made a remarkable attempt to explore the link between EAT to renal disease, the body of evidence that examines the relation of this particular fat depot to renal disease seems to be insufficient and in its early days, requiring further analysis and clinical studies.

Author Response

Response to Reviewer one comments,

We want to thank the reviewers for evaluating the manuscript and providing with valuable comments and suggestions.

Please find our responses to the reviewer one as below,

  1. In this paper Aeddula et at review in detail the exiting knowledge on epicardial adipose tissue (EAT) and renal disease. Although the manuscript is clear and well organized with a complete description on the role of EAT on cardiac metabolism and disease; the link that they try to achieve between EAT and renal disease seems to be at best not well established. The authors propose the measurement of the EAT thickness as an early marker of cardiovascular disease (previously suggested in Russo et al J Nephrol. 2018) and discuss different strategies to reduce the EAT in patients with renal disease such as life-style changes that involve low calorie diets and anaerobic exercise. This course of treatment will impact not only EAT but all the fat depots, therefore, the particular beneficial effect of it cannot be attribute to an EAT reduction. 

  2. In conclusion, although the authors made a remarkable attempt to explore the link between EAT to renal disease, the body of evidence that examines the relation of this particular fat depot to renal disease seems to be insufficient and in its early days, requiring further analysis and clinical studies.

Response:

The cardiovascular morbidity and mortality in renal disease are high across the globe, and there is a lack of large-scale studies, the paucity of concrete evidence and societal guidelines to detect early CVD in this group of patients. Our effort was to perform a thorough clinical review on the existing evidence and knowledge on the EAT in patients with renal disease, as a reliable early noninvasive biomarker and indicator for CVD.

We want to thank the reviewer’s comment, that Russo and his collaborators from Italy have proposed in 2018, that measurement of EAT as an early marker of cardiovascular disease. We have discussed and included his comments in our original manuscript (Citation number 69).

We agree with the reviewer’s comments that the current evidence on various strategies available to reduce the EAT in patients with renal disease could impact all the fat depots elsewhere in the body. But multiple studies have found a positive association between epicardial fat volume and coronary and carotid artery atherosclerosis independent of other cardiovascular risk factors (1,2,3). Moreover, Cordeiro et al. showed that EAT volume in CKD (stage 3-5) is a better predictor of the risk of cardiovascular events than the abdominal visceral adipose tissue, (4).

We agree and accede that the cause-effect relationship between EAT volume, cardiovascular risk, and renal disease has not been established and concur with the comment that further large-scale clinical trials are needed to establish the link. We have modified the conclusion of the review, to reflect the reviewer's suggestions and comments and hope the reviewer finds it satisfactory.

References:

1. Mazurek T, Zhang L, Zalewski A, Mannion JD, Diehl JT, Arafat H et al. (2003) Human epicardial adipose tissue is a source of inflammatory mediators. Circulation 108(20):2460–2466

2. Konishi M, Sugiyama S, Sugamura K, Nozaki T, Ohba K, Matsubara J et al. (2010) Association of pericardial fat accumulation rather than abdominal obesity with coronary atherosclerotic plaque formation in patients with suspected coronary artery disease. Atherosclerosis 209(2):573–578

3. Bos D, Shahzad R, van Walsum T, van Vliet LJ, Franco OH, Hofman A et al. (2015) Epicardial fat volume is related to atherosclerotic calcification in multiple vessel beds. Eur Heart J Cardiovasc Imaging 16:1264–1269

4. Cordeiro AC, Amparo FC, Oliveira MAC, Amodeo C, Smanio P, Pinto IMF et al. (2015) Epicardial fat accumulation cardiometabolic profile and cardiovascular events in patients with stages 3–5 chronic kidney disease. J Intern Med 278(1):77–87.

Reviewer 2 Report

Current review by Aeddula et al. presents a cumulative account of epicardial adipose tissue and chronic kidney diseases. The review is detailed, interesting and well written. This reviewer have some minor points as following.

1. In some of the sentences throughout the manuscript, there are over use of 'and' like line 107-110.

2. The authors could add some details explaining/ proposing role of EAT in controlling systemic metabolism to control CKD, as EAT represents a small portion of adipose mass. Does EAT levels induce/ take part in CKD development or high blood pressure due to CKD make changes in EAT levels along with other cardiac changes.

3. There are some recent studies in this same subject area which can be added in this review.  like PMID:30674009

Author Response

Response to Reviewer-two comments,

We want to thank the reviewers for evaluating the manuscript and providing with valuable comments and suggestions.

Please find our responses to the reviewer two as below,

2. Current review by Aeddula et al. presents a cumulative account of epicardial adipose tissue and chronic kidney diseases. The review is detailed, interesting and well written. This reviewer have some minor points as following.

1. In some of the sentences throughout the manuscript, there are over use of 'and' like line 107-110.

2. The authors could add some details explaining/ proposing role of EAT in controlling systemic metabolism to control CKD, as EAT represents a small portion of adipose mass. Does EAT levels induce/ take part in CKD development or high blood pressure due to CKD make changes in EAT levels along with other cardiac changes.

3. There are some recent studies in this same subject area which can be added in this review.  like PMID:30674009

Response:

We want to thank the reviewer for his comment that the review is detailed, interesting and well written.

1.       We want to apologize for the overuse of the word ‘and’ in the manuscript and since we have modified the review and made significant changes.

2.       We appreciate the reviewer’s suggestion on including the latest evidence. We have added the study by Reinhardt et al., (published 01/16/2019) on the role of EAT in predicting the decreased renal function and coronary artery calcification in youth and early adult-onset type 2 DM. We have modified the manuscript, and discussion is included under the clinical significance of EAT in renal disease. (citation no 86).

3.       Erdogan et al., have studied the EAT thickness in 150 patients with hypertension and found that EAT can affect the blood pressure by several mechanisms,(1).

4.       We have not found any conclusive evidence in the literature to show that EAT levels can induce CKD.

We hope that the reviewer finds the manuscript modifications satisfactory.

Thank you

References:

1.        Erdogan G, Belen E, Sungur MA, Sungur A, Yaylak B, Güngör B, et al. Assessment of epicardial adipose tissue thickness in patients with resistant hypertension. Blood Press Monit [Internet]. 2016 Feb;21(1):16–20. Available from: http://www.ncbi.nlm.nih.gov/pubmed/26317386

Round  2

Reviewer 1 Report

In this paper Aeddula et at review in detail the exiting knowledge on epicardial adipose tissue (EAT) and renal disease. Although I still think that the link that between EAT and renal disease is in its early days, I believe that the current version of the review (with the changes made to the conclusion) now reflects the potential of EAT as a marker and points out the work that still needs to be done. Finally, I would advice the authors to read thoroughly the manuscript again since there is still some editing to be done (e.g. line 134 physiological instead of physiologic; line 160 be instead of me).

Author Response

Response to Reviewer one( round two) comments,

We want to thank the reviewers for evaluating the manuscript and providing with valuable comments and suggestions.

Please find our responses to the reviewer one as below,

1. In this paper Aeddula et at review in detail the exiting knowledge on epicardial adipose tissue (EAT) and renal disease. Although I still think that the link that between EAT and renal disease is in its early days, I believe that the current version of the review (with the changes made to the conclusion) now reflects the potential of EAT as a marker and points out the work that still needs to be done. Finally, I would advice the authors to read thoroughly the manuscript again since there is still some editing to be done (e.g. line 134 physiological instead of physiologic; line 160 be instead of me).

Response:

We appreciate the reviewer one’s comments. We have made further editing and modifications to the manuscript including the line 134,160 and others. Please do accept our sincere apologies for the inconvenience and hope the reviewer finds the revised document satisfactory.

Thank you for your time and consideration.  We look forward to hearing from you.

Sincerely,

Narothama Reddy Aeddula, MD

Division of Nephrology, Deaconess Health System

Dr.anreddy@gmail.com